

# Testing intra-species variation in allocation to growth and defense in rubber tree (*Hevea brasiliensis*)

Kanin Rungwattana[1], Poonpipope Kasemsap[2,3], Thitaporn Phumichai[4], Ratchanee Rattanawong[5] and Peter Hietz[6]

[1] Department of Botany, Faculty of Science, Kasetsart University, Bangkok, Thailand
[2] Hevea Research Platform in Partnership, DORAS Center, Kasetsart University, Bangkok, Thailand
[3] Department of Horticulture, Faculty of Agriculture, Kasetsart University, Bangkok, Thailand
[4] Rubber Authority of Thailand, Bangkok, Thailand
[5] Nong Khai Rubber Research Center, Rubber Authority of Thailand, Nong Khai, Thailand
[6] Institute of Botany, Universität für Bodenkultur Wien, Vienna, Austria

## ABSTRACT

**Background:** Plants allocate resources to growth, defense, and stress resistance, and resource availability can affect the balance between these allocations. Allocation patterns are well-known to differ among species, but what controls possible intra-specific trade-offs and if variation in growth *vs.* defense potentially evolves in adaptation to resource availability.

**Methods:** We measured growth and defense in a provenance trial of rubber trees (*Hevea brasiliensis*) with clones originating from the Amazon basin. To test hypotheses on the allocation to growth *vs.* defense, we relate biomass growth and latex production to wood and leaf traits, to climate and soil variables from the location of origin, and to the genetic relatedness of the *Hevea* clones.

**Results:** Contrary to expectations, there was no trade-off between growth and defense, but latex yield and biomass growth were positively correlated, and both increased with tree size. The absence of a trade-off may be attributed to the high resource availability in a plantation, allowing trees to allocate resources to both growth and defense. Growth was weakly correlated with leaf traits, such as leaf mass per area, intrinsic water use efficiency, and leaf nitrogen content, but the relative investment in growth *vs.* defense was not associated with specific traits or environmental variables. Wood and leaf traits showed clinal correlations to the rainfall and soil variables of the places of origin. These traits exhibited strong phylogenetic signals, highlighting the role of genetic factors in trait variation and adaptation. The study provides insights into the interplay between resource allocation, environmental adaptations, and genetic factors in trees. However, the underlying drivers for the high variation of latex production in one of the commercially most important tree species remains unexplained.

Corresponding author
Kanin Rungwattana,
kanin.run@ku.th

## INTRODUCTION

Plants require resources to support growth, reproduction, defense, and stress resistance. Resources such as energy, nutrients, and water are often limited in natural habitats. Growth demands a substantial influx of carbon obtained through photosynthesis for the metabolic processes of meristematic tissues. Also the synthesis of defensive compounds against herbivores (*Mithöfer & Boland, 2012*; *Müller et al., 2024*; *Younkin et al., 2024*), pathogens (*Ullah et al., 2017*; *Akbar et al., 2023*; *Anjali et al., 2023*), and for detoxification (*Neilson et al., 2013*; *Kidwai, Ahmad & Chakrabarty, 2020*; *Du et al., 2024*) is reliant on this carbon supply (*Garcia et al., 2021*). Therefore, when plants have a limited pool of resources, such as carbon and nutrients, there is a potential trade-off between promoting growth and enhancing defense mechanisms (*Herms & Mattson, 1992*; *Figueroa-Macías et al., 2021*; *He, Webster & He, 2022*). Such a trade-off will result in trees that grow quickly tending to have weak defenses, while those with strong defenses tend to grow slowly (*Fichtner et al., 2017*; *Züst & Agrawal, 2017*; *Zhu et al., 2018*). When plants allocate more resources towards growth, this may therefore come at the expense of defense, making the plant more vulnerable to attacks from herbivores, pathogens, and also to abiotic stresses. Conversely, when plants prioritize defense, they produce compounds such as secondary metabolites, toxins, and structural barriers that deter herbivores and pathogens or may also mitigate the effects of environmental stressors. Allocating resources to defense mechanisms may limit the resources available for growth, potentially slowing down developmental processes. Various hypotheses try to explain how plants balance the investment in either growth or defense. The resource availability hypothesis proposes that the availability of resources is the major determinant of plant defensive investment (*Coley, Bryant & Chapin, 1985*). When resources are limited, slow-growing species with high investment in defense are favored over fast-growing species. Consequently, plants inhabiting high-stress environments are more likely to have developed elevated levels of defense mechanisms compared to those in low-stress environments (*Coley, Bryant & Chapin, 1985*; *Grime, 2006*). The carbon-nutrient balance hypothesis explains the pattern of plant defense with the C/N ratio in plant tissues (*Bryant, Chapin & Klein, 1983*). Plants growing in low-nutrient environment are more likely to produce carbon-based defensive metabolites (*e.g.*, isoprene, terpenes, phenolics), whereas plants under low carbon availability tend to use nitrogen-based metabolites (*e.g.*, alkaloids, nonprotein amino acids, cyanogenic compounds) (*Hattas, Scogings & Julkunen-Tiitto, 2017*). The growth-differentiation balance hypothesis encompasses and extends the carbon-nutrient balance hypothesis by proposing that any environmental factor that limits growth to a greater extent than it limits photosynthesis will increase the available resources allocated to carbon-based secondary metabolism (*Herms & Mattson, 1992*; *Glynn et al., 2007*). Thereby, plants facing high-stress conditions tend to invest more in producing structural or chemical defenses, while plants in low-stress conditions prioritize growth over defense.

Resources that potentially limit tree growth in closed tropical forest include light, mineral nutrients, and water (*Kitajima & Poorter, 2008*). These factors not only influence the allocation of resources towards primary and secondary metabolic processes but also
play a crucial role in determining the trade-off between growth and defense, which are both vital for plant fitness and survival (*Herms & Mattson, 1992*). The C/N ratio in plants can be influenced by variations in light intensity (*Zavala & Ravetta, 2001*; *Szabó et al., 2020*). Low light constrains carbon assimilation, leading to a decrease in carbon-based metabolites. Under these conditions the limited carbon available is primarily allocated towards growth, rather than secondary compounds, while the production of nitrogen-based secondary metabolites is adequately supported if nitrogen supply is good relative to light (*Xiao et al., 2022*). Conversely, high light intensity drives photosynthesis, resulting in higher production of carbon-based secondary metabolites (*Bryant, Chapin & Klein, 1983*; *Pant, Pandey & Dall'Acqua, 2021*). Growth processes are generally more sensitive to nutrient or water supply as opposed to photosynthesis (*Körner, 1991*; *Luxmoore, 1991*). Moderate nutrient deficiency and moderate drought thereby affect source-sink interaction. For example, when water limitation reduces growth more than photosynthesis under moderate drought, plants accumulate carbohydrates leading to an increased C/N ratio within the plant (*Bradford & Hsiao, 1982*). In such conditions, carbohydrates not required for growth can be allocated to carbon-based secondary metabolites with little or no trade-off between growth and defense (*Horner, 1990*; *Zhang et al., 2017*; *Cheng et al., 2018*). Consequently, plants facing sink limitation due to stress tend to display higher resistance to external constraints compared to plants that encounter no environmental limitations in their growth (*Herms & Mattson, 1992*). *Körner (1998)* hypothesized that trees at the treeline are particularly sink limited due to low temperatures during the growing season, and also referred to the growth limitation hypothesis. In subsequent years, this hypothesis had been tested and renamed the sink limitation hypothesis (*Susiluoto et al., 2007*). However, their findings presented conflicting evidence, suggesting that the photosynthesis of trees at the treeline may not be limited by sinks. This is evidenced by the notable increase in growth of the apical meristem after debudding, while transpiration rates remained unchanged.

An evolutionary trade-off between growth and defense has been widely observed across species (*Endara & Coley, 2011*). However, there is a significant gap in our understanding of these patterns within species, specifically if there is a trade-off between investment in growth *vs.* defense that differs among populations and how this might be driven by adapted to different environments. Many species with large geographical ranges occur across substantial gradients of environmental factors. As local populations adapt to these (*Louthan, Doak & Angert, 2015*) and drivers of evolution should also affect populations within a species. The resource availability hypothesis suggests that environmental factors affect resource acquisition and allocation. As a result, the associations between growth and defense traits can vary in strength and direction across different populations (*Agrawal, 2011*; *Hahn & Maron, 2016*). This phenomenon was observed as gradual changes, known as clines, in plant growth and defense traits along environmental gradients, such as relatively small latitudinal shifts (*Petit & Hampe, 2006*) or changes in elevation over a few hundred meters (*Pratt & Mooney, 2013*). Plant populations evolve diverse traits adapting to local environments (*Fine et al., 2006*), leading to genetic variation among populations. Whether traits have evolved in response to environmental constraints can be tested in

common garden experiments where trees from diverse climates are grown in a uniform location to reduce phenotypic variation in adaptation to the growth environment (*George et al., 2017*; *Vázquez-González et al., 2020*; *Cope et al., 2021*). If populations have locally adapted to resource availability (*Savolainen, Pyhäjärvi & Knürr, 2007*) this might affect resource allocation to growth and defense and a potential trade-off between these two. While growth in biomass is straightforward to standardize, trees can be defended by a range of secondary compounds, which complicates a quantitative comparison of the overall investment in defense unless there is little variation in their chemical nature. In pines polyphenols are the major defensive compounds and the polyphenol concentration in *Pinus pinaster* leaves was negatively correlated with tree growth, but only under phosphorus-limitation (*Sampedro, Moreira & Zas, 2011*). While this greenhouse experiment with half-sib families found allocation also under genetic control, it was not tested if the differences observed reflect adaptations to different resource availabilities at the location of origin of the mother trees. A common garden experiment with *Asclepias syriaca* populations from a wide latitudinal range—related to annual precipitation—found clinal variations in growth opposing clines in latex production (*Woods et al., 2012*).

Latex can contain a range of compounds, is the primary defense mechanisms in many plants and can be quantified and has thus be suggested as a model to study mechanisms, ecology, and evolution of plant defense against herbivory (*Agrawal & Konno, 2009*). So far, the ecology and evolution of latex production has been studied mainly in milkweeds (*Asclepias*; *Agrawal, 2011*). In rubber trees (*Hevea brasiliensis*) the prime defense compound in latex is an isoprene polymer (natural rubber) that includes no nutrients, is produced in large quantities, and is harvested and quantified in a standardized way, which makes it perhaps the best model to investigate the growth-defense trade-off in a tropical tree. Most commercially cultivated rubber trees are clones derived from the narrow gene pool collected in 1876 (*Onokpise, 2004*). To broaden the genetic base for natural rubber production, wild *Hevea* seeds had been collected in 1981 from native populations in the Amazon basin by the International Rubber Research and Development Board (known as the IRRDB'81 collection) and were subsequently distributed as clones to many countries in Southeast Asia and Western Africa. Using a provenance trial based on this collection planted in Thailand, wood and leaf traits have previously been analyzed by *Rungwattana et al. (2018)*, who found leaf mass per area (LMA), leaf $\delta^{13}$C and leaf N per area positively correlated to rainfall during the dry seasons (*i.e.*, negatively to dry season intensity) of the place of origin of the clones, and marginally significant ($p < 0.1$) correlations with wood traits. In the present study, we use additional data on wood traits, tree growth and latex production of the same provenance trial to test the resource availability hypothesis on intraspecific variation in allocation to growth and defense, and ask: (1) Is there a trade-off between investment in growth and latex yield? (2) Which traits control the investment in growth and defense? (3) Is investment in growth or defense related to clinal variation and potential genetic adaptations to water or nutrient availability? Comparing trees in a common garden experiment we are testing for genetically controlled differences and whether these are related to evolutionary adaptations to the local environment in Brazil the

genotypes originate from. We also test for a phylogenetic signal in the traits using a phylogenetic tree based on single nucleotide polymorphisms (SNPs).

## MATERIALS AND METHODS

### Plant materials and study site

The Brazilian rubber trees sampled are part of the International Rubber Research and Development Board IRRDB'81 collection. Parent materials had originally been sampled from wild populations in forests in the Amazon basin in Brazil from different municipalities in the states of Acre, Rondônia and Mato Grosso with a wide range of rainfall. Annual mean temperature at the locations of origin ranges between 21.4 °C and 26 °C, annual precipitation between 1,501 and 2,254 mm, and precipitation in the driest quarter between 19 and 199 mm (obtained from www.worldclim.org). Organic carbon density at the locations of origin ranges between 201.7 and 296.3 g dm$^{-3}$, bulk density between 115.0 and 131.7 cg cm$^{-3}$, sand content between 188.0 and 664.7 g kg$^{-1}$, cation exchange capacity between 81.3 and 182.3 mmol kg$^{-1}$, soil nitrogen content between 126.7 and 233.7 cg kg$^{-1}$, organic carbon content between 148.3 and 436.7 dg kg$^{-1}$, and pH between 4.2 and 5.3 (obtained from www.soilgrids.org). In 1994, clones from these parent trees were planted with five replicates per clone at Nong Khai Rubber Research Center, Thailand (18°09′30″ N, 103°09′31″ E). Trees were planted with a spacing of 3 × 7 m, and each tree was fertilized twice a year with 500 g 30:5:18 (N:P:K) fertilizer. Mean annual rainfall at the trial site is 1,671 mm and average temperature is 27.5 °C. The rainfall during the dry season between November and April is normally less than 100 mm per month, and the total of the driest quarter between December and February is approximately 60 mm (*Thai Meteorological Department, 2017*). The climate and soil variables of the locations of origin in the Amazon basin in Brazil and the trial site in Thailand are presented in Table 1.

### Sample and data collections

Latex was sampled monthly between May and January, but not during dry season when leaves are shed. To sample latex, narrow incisions about 0.2 cm in width and 0.7 cm in depth that slope downwards approximately 30° and run around half the stem circumference were cut into the bark, starting at 1.5 m above the stem base. In each tapping, a strip of bark was removed on the same cut every other day. Latex was collected from a recipient every other day. When the successive incisions reached approximately 20 cm above the stem base, latex collection started at 1.5 m at the other half of the stem and proceeded downwards, typically over five years. At this rubber plantation no stimulant to increase latex production had been used. The latex produced by all trees per clone was pooled and converted to grams/tree/tapping (g/t/t). Tree circumference was measured at 1.7 m above the stem base in March every year between 2014 and 2019. Tree diameter (D) was calculated from circumference/π, and growth rate was calculated as diameter increment per year. We also calculated the biomass growth rate as the change in the cross-sectional area between two years multiplied by the wood density. While this does not equate to the biomass growth of the entire tree, this is also the case for the latex collected and both measures would scale per unit stem length.

**Table 1 Comparison of geographical distribution, climate, and soil characteristics among *Hevea brasiliensis* populations: Amazon basin *vs.* provenance trial in Nong Khai, Thailand.**

| State | Municipality | Lat | Lon | Climate | | | | Soil | | | | | | |
|---|---|---|---|---|---|---|---|---|---|---|---|---|---|---|
| | | | | T | P | Pcv | Pq | OCD | Bulkd | Sand | CEC | N | SOC | pH |
| Acre | Assis Brasil | −10.89 | −69.58 | 24.6 | 1,640 | 61 | 66 | 221.7 | 122.3 | 375.0 | 107.0 | 148.0 | 148.3 | 4.8 |
| Acre | Feijo | −8.16 | −70.35 | 25.9 | 2,205 | 52 | 152 | 296.3 | 119.7 | 188.0 | 182.3 | 233.7 | 436.7 | 4.6 |
| Acre | Ortet | −7.86 | −70.68 | 25.8 | 2,254 | 47 | 199 | 252.7 | 121.0 | 312.7 | 132.0 | 185.7 | 228.7 | 4.7 |
| Acre | Sena Madureira | −9.07 | −68.66 | 24.8 | 2,017 | 53 | 140 | 272.7 | 121.3 | 203.7 | 167.3 | 203.7 | 254.7 | 4.7 |
| Mato Grosso | Aracatuba | −13.71 | −54.91 | 23.6 | 1,721 | 76 | 33 | 211.0 | 126.0 | 584.7 | 81.3 | 127.7 | 176.7 | 5.3 |
| Mato Grosso | Cartriquaçu | −9.86 | −58.41 | 24.8 | 1,980 | 73 | 46 | 216.3 | 118.7 | 471.7 | 125.0 | 144.3 | 174.3 | 4.5 |
| Mato Grosso | Itauba | −11.06 | −55.28 | 25.4 | 1,885 | 75 | 19 | 225.0 | 125.3 | 484.3 | 108.3 | 126.7 | 151.3 | 4.7 |
| Mato Grosso | Villa Bela | −12.97 | −60.11 | 21.4 | 1,943 | 70 | 57 | 201.7 | 123.3 | 664.7 | 93.7 | 142.0 | 170.7 | 4.8 |
| Rondonia | Ariquemenes | −9.91 | −63.04 | 25.4 | 2,184 | 64 | 62 | 258.7 | 122.3 | 461.3 | 124.0 | 150.7 | 207.0 | 4.7 |
| Rondonia | Calama | −8.76 | −63.90 | 26.0 | 2,096 | 60 | 100 | 232.3 | 115.0 | 358.3 | 134.0 | 150.3 | 184.3 | 4.2 |
| Rondonia | Costa Marques | −12.45 | −64.23 | 26.0 | 1,501 | 69 | 35 | 243.0 | 126.7 | 504.0 | 87.7 | 158.3 | 165.7 | 4.8 |
| Rondonia | Jaru | −10.44 | −62.47 | 24.9 | 1,989 | 70 | 47 | 238.7 | 126.7 | 447.7 | 148.3 | 146.3 | 173.3 | 4.7 |
| Rondonia | Jiparana | −10.89 | −61.95 | 24.5 | 1,951 | 70 | 51 | 278.7 | 123.7 | 461.7 | 132.0 | 187.7 | 181.7 | 4.6 |
| Rondonia | Ouro Preto | −10.75 | −62.22 | 24.4 | 1,924 | 72 | 42 | 280.3 | 120.3 | 516.7 | 144.0 | 182.3 | 275.7 | 4.7 |
| Rondonia | Pimenta Bueno | −11.67 | −61.19 | 23.8 | 1,922 | 70 | 51 | 231.7 | 131.7 | 391.3 | 124.7 | 160.3 | 158.7 | 4.9 |
| Provenance trial in Nong Khai, Thailand | | 103.09 | 18.09 | 27.5 | 1,671 | 101 | 60 | 246.3 | 132.3 | 375.0 | 122.7 | 163.7 | 154.7 | 5.5 |

Note:
Lat, latitude; lon, longitude T, annual mean temperature (°C); P, annual precipitation (mm); Pcv, precipitation seasonality (CV); Pq, precipitation of driest quarter (mm); OCD, organic carbon density (g $dm^{-3}$); bulkd, bulk density (cg $cm^{-3}$); sand, sand content (g $kg^{-1}$); CEC, cation exchange capacity (mmol $kg^{-1}$); N, nitrogen content (cg $kg^{-1}$); SOC, organic carbon content (dg $kg^{-1}$); and pH, pH water. Trees grown in the provenance trial were fertilized twice a year with 500 g 30:5:18 (N:P:K) fertilizer per tree.

Leaf traits had been obtained previously in 2013 (see *Kanpanon et al., 2017*; *Rungwattana et al., 2018* for details). For the analysis of trait-coordination among clones and the relationship between traits and resource availability at the place of origin we used the mean values for each of 49 clones for which we had 3–5 replicates per clone plus 101 additional clones with one replicate per clone. Sun-exposed leaves were collected to measure LMA, leaf C and N content, and the carbon isotope signal ($\delta^{13}C$). LMA is the ratio between leaf dry mass and leaf area (*Poorter et al., 2009*) and reflects acquisitive *vs.* conservative resource use for low and high values, respectively. Leaf $\delta^{13}C$ was used to calculate intrinsic water use efficiency (iWUE), which is $CO_2$ uptake per stomatal conductance ($A/g_s$) following *Farquhar, Ehleringer & Hubick (1989)*.

Wood samples were collected in 2017 with a 5.15-mm-diameter increment borer (SUUNTO, Vantaa, Finland) at a height of c. 1.95 m and cut to 3 cm length from the cambium (*Rungwattana et al., 2018*). Wood samples were collected from 153 clones in total; 70 clones were sampled with 3–5 replicates to test for within-clone variation, and an additional 83 clones with one replicate per clone We prepared 30-μm-thick transverse sections and took images with a DM5500B transmission light microscope with a DMC2900 camera (Leica, Germany) with a resolution of 864 pixel/mm and covering an area of c. 65 $mm^2$ (see *Rungwattana et al., 2018* for details).

Average vessel area (VA), vessel densities (VD), vessel area fraction (VF), theoretical hydraulic conductivity (Kp), and wood density (WD) had previously been measured (*Rungwattana et al., 2018*). We additionally measured the vessel grouping index (VGI) and theoretical implosion resistance $(t/b)^2$ (*Hacke et al., 2001*). VGI is the average number of vessels belonging to one group (vessels in direct contact) or the total number of vessels divided by the total number of vessel groupings (*Carlquist, 2013*). A VGI of 1 indicates exclusively solitary vessels, the higher the index, the greater degree of vessel grouping (*Scholz et al., 2013*). For $(t/b)^2$, we measured maximum diameter of the vessel (b) and double wall thickness of vessel contact walls (t) for c. 10 vessel pairs per section. We used ROXAS (https://www.wsl.ch/en/projects/roxas-x/) plugged in with specific algorithm to automatically detect conduits and calculate VGI of individual sections (*von Arx, Kueffer & Fonti, 2013*). The b and t were not measured on the saved images but on higher-resolution live images with the same microscope.

To explore possible trade-offs between growth, defense, and stress (drought) resistance, we distinguished four categories of traits. (1) Investment in growth, derived from tree biomass growth; (2) investment in defense, derived from latex yield; (3) traits related to the leaf economics spectrum *i.e.*, leaf mass per area (LMA), leaf carbon and nitrogen content (leaf C and N). (4) traits related to water relations, *i.e.*, intrinsic water use efficiency (iWUE), theoretical hydraulic efficiency (Kp), vessel density (VD), vessel area (VA), vessel area fraction (VF), vessel grouping index (VGI), and theoretical vessel implosive resistance $(t/b)^2$.

## Data analysis

We first calculated simple regressions between biomass growth and latex produced per tree to test for a trade-off that would be seen in a negative correlation. Given that both increased with tree size (Fig. 1), we included tree size in the model to test if yield is correlated with growth independent of tree size. Latex is collected by incisions around half the stem circumference, so longer cuts are made in thicker trees and will produce more latex. Therefore, to reduce the effect of tree size for the subsequent analyses we scaled growth and latex production to diameter and hereafter refer to "growth" as biomass growth/D and "latex yield" as the average amount of latex produced per tapping/D.

We tested for a trade-off between growth and defense in two alternative ways. First, we calculated a linear regression between growth and latex yield (both scaled to tree size) to test if trees that invest more in growth, invest less in defense. Since this was not the case, we considered that trees might differ in the total carbon resource available for other reasons so that trees with more C could invest more in growth as well as defense. We therefore asked if the relative investment in growth *vs.* defense differs and is related to traits or environmental variables from the place of origin. For this, we scaled the biomass increment of each clone by dividing by the maximum increment of all clones (Grel), and likewise for latex yield (Lrel). We then calculated the proportional investment in growth (pG) as Grel/(Grel + Lrel).

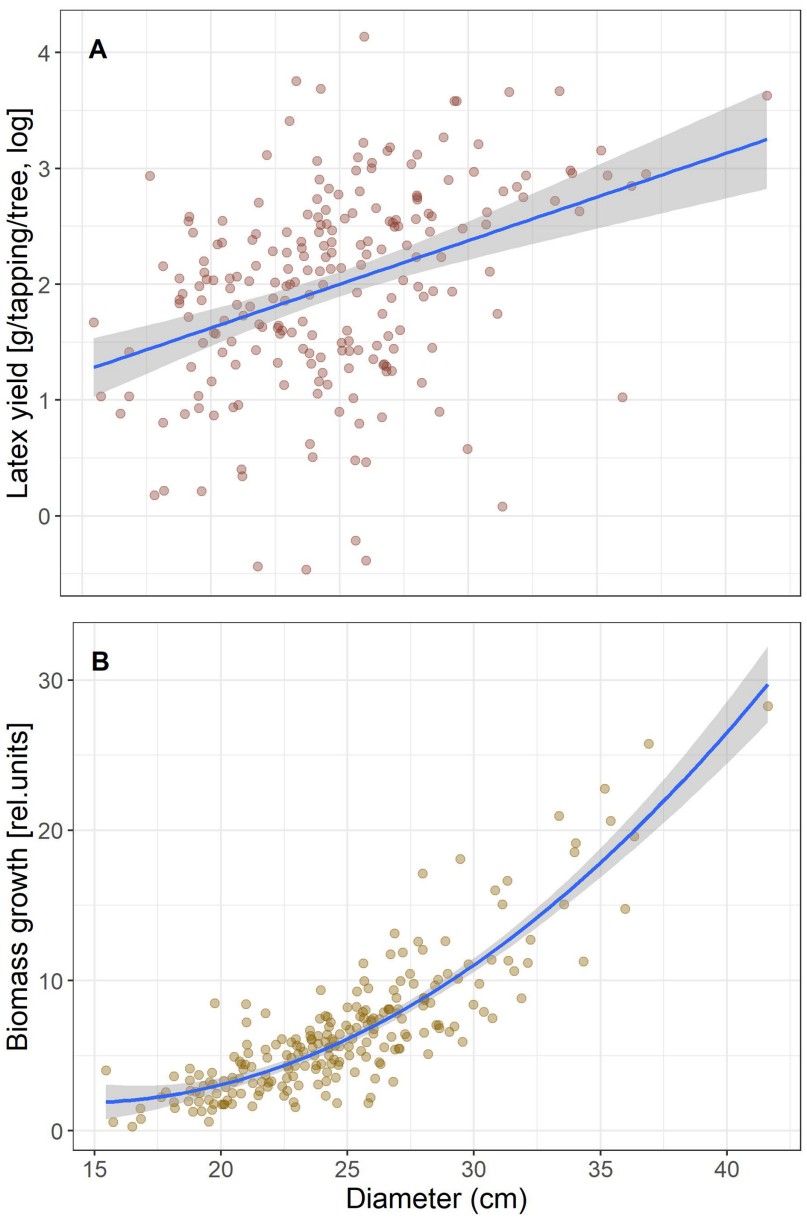

**Figure 1 Correlations of latex production (A) and tree diameter growth (B) with tree size (Diameter).**

We then tested whether growth, latex yield or the proportional investment in growth (pG) are related to any single traits, as well as with the first two principal components of analysis (PCA) of traits in pairwise linear regressions.

The genetic relationships of the clones were shown in a phylogenetic tree based on 1,820 single nucleotide polymorphisms (SNPs, *Chanroj et al., 2017*). We used this tree to calculate the phylogenetic signals for growth, latex yield, the relative investment in growth, and other traits using Blomberg's K (*Blomberg, Garland & Ives, 2003*), calulated with R library phytools (*Revell, 2012*). To test if a possible relationship between growth, latex yield and traits is affected by the phylogenetic relationship between clones, we calculated

pairwise regressions with traits and the PCA axes as above, but using phylogenetic generalized linear models with pgls in R library caper (*Orme et al., 2018*) with maximum likelihood branch length transformation and default settings.

To test if growth, latex yield or traits are related to the environment of the clones' location of origin, we correlated these variables with bioclimate parameters at locations of origin (mean annual temperature: MAT, mean annual precipitation: MAP and precipitation in the driest quarter: Pq) from WorldClim (www.worldclim.org) and soil variables (organic carbon density: OCD, bulk density: bulkD, sand content: Sand, cation exchange capacity; CEC, nitrogen content: soilN, organic carbon content: SOC, and pH) obtained from SoilGrids (www.soilgrids.org). Since soil parameter can vary locally at small scales, the sampling position of the clones in Brazil is not very exact, and we consider these to present local populations, we extracted averages for all soil parameters for areas of 10 × 10 km around the given sampling positions. For these pairwise correlations we used mixed-effect models with "location" as a random effect using R package lme4 (*Bates et al., 2015*) because several clones had been obtained from each location and thus variables related to location are pseudo-replicates. We did not include the phylogenetic relationship in the analysis of local variables because the point here was to test if there are related, and not if this relationship is phylogenetically constrained.

# RESULTS

## Is there a trade-off between investment in growth and latex yield?
Tree growth and latex production per tree are positively correlated ($r^2 = 0.11$, $p = 7.6$ E−7), and both scale with tree size (Fig. 1). When tree size and tree growth were included in the model explaining latex production, latex production was affected by tree size ($p = 0.0024$), but not by growth ($p = 0.83$). If latex yield and biomass growth were expressed per stem diameter, the correlation between biomass growth and latex yield was still positive and significant, but weak ($r^2 = 0.021$, $p = 0.031$). Thus, we found no trade-off, which would have been indicated by a significant negative correlation, between the investment in growth and defense for the Brazilian rubber trees.

## What controls investment in growth *vs.* defense?
We then asked whether growth, defense or the relative investment in growth *vs.* defense are correlated with any other traits or a combination of traits as seen in a PCA. While biomass growth, scaled to tree diameter, correlated with WD, LMA, iWUE and leaf N (Table 2), these correlations were weak ($r^2 < 0.07$). Latex yield was marginally related to VGI and the relative investment in growth *vs.* latex (pG) was marginally related to VGI and VD. The significance in phylogenetic regressions were similar to pairwise models, but iWUE was not related to growth. In the PCA, leaf traits (LMA, iWUE, and leaf N content, but not leaf C content) scaled strongly along PC1, together with tree growth (Fig. 2A). Wood traits scaled mostly along PC2, and latex yield had very little relation with PC1 or PC2. Scores in the phylogenetic PCA were similar (Fig. 2B, note that mirroring along the axes is not relevant in the comparison of PCAs), but WD scaled stronger along PC1.

**Table 2 Significance of correlations between functional traits and growth, latex yield and the relative investment in growth (pG) calculated with pairwise linear regressions and phylogenetic regressions.**

| Trait | Pairwise | | | Phylogenetic | | |
|---|---|---|---|---|---|---|
| | Growth | Latex | pG | Growth | Latex | pG |
| Wall | 0.995 | 0.634 | 0.819 | 0.821 | 0.270 | 0.255 |
| tb2 | 0.291 | 0.141 | 0.882 | 0.318 | 0.440 | 0.991 |
| VGI | 0.882 | **0.051** | **0.069** | 0.383 | **0.052** | 0.238 |
| WD | **0.002** | 0.748 | 0.202 | **0.013** | 0.708 | 0.116 |
| VA | 0.843 | 0.824 | 0.980 | 0.892 | 0.762 | 0.807 |
| VD | 0.106 | 0.995 | **0.072** | 0.539 | 0.300 | **0.075** |
| VF | 0.125 | 0.351 | 0.261 | 0.737 | 0.996 | 0.398 |
| Kp | 0.117 | 0.736 | 0.141 | 0.588 | 0.608 | 0.206 |
| LMA | **0.001** | 0.931 | 0.434 | **0.005** | 0.381 | 0.602 |
| iWUE | **0.019** | 0.291 | 0.923 | 0.313 | 0.172 | 0.579 |
| N | **0.011** | 0.401 | 0.823 | **0.038** | 0.291 | 0.757 |
| C | 0.816 | 0.529 | 0.605 | 0.750 | 0.443 | 0.563 |

**Note:**
Significant and marginally significant ($p < 0.1$) $p$-values are printed in bold.

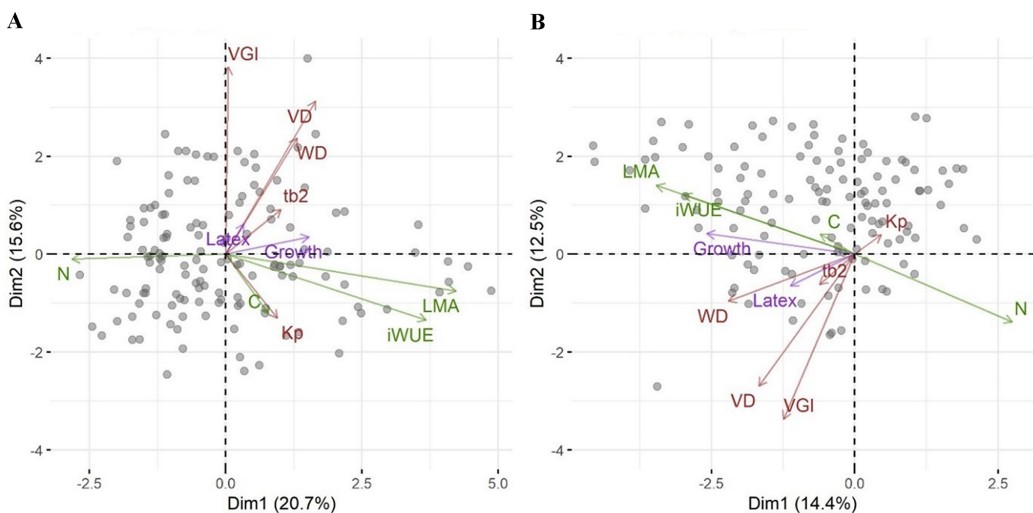

**Figure 2 Principal component analysis of traits for the Brazilian rubber clones.** A standard PCA based on scaled trait values (A), and a phylogenetic PCA based on scaled trait values and the genetic relationship among clones (B). Each data point indicates one clone. Arrows represent the ordination of traits along two principal component axes with green for leaf traits, brown for wood traits, and magenta for investment in growth and defense. Abbreviations are: pG, relative investment in growth *vs.* latex, wall, wood vessel wall thickness, VGI, vessel grouping index, WD, wood density, VA, vessel area, VD, vessel density, VF, vessel fraction, Kp, theoretical hydraulic conductivity, LMA, leaf mass per area, iWUE, intrinsic water use efficiency, N, leaf nitrogen concentration, C, leaf carbon concentration.

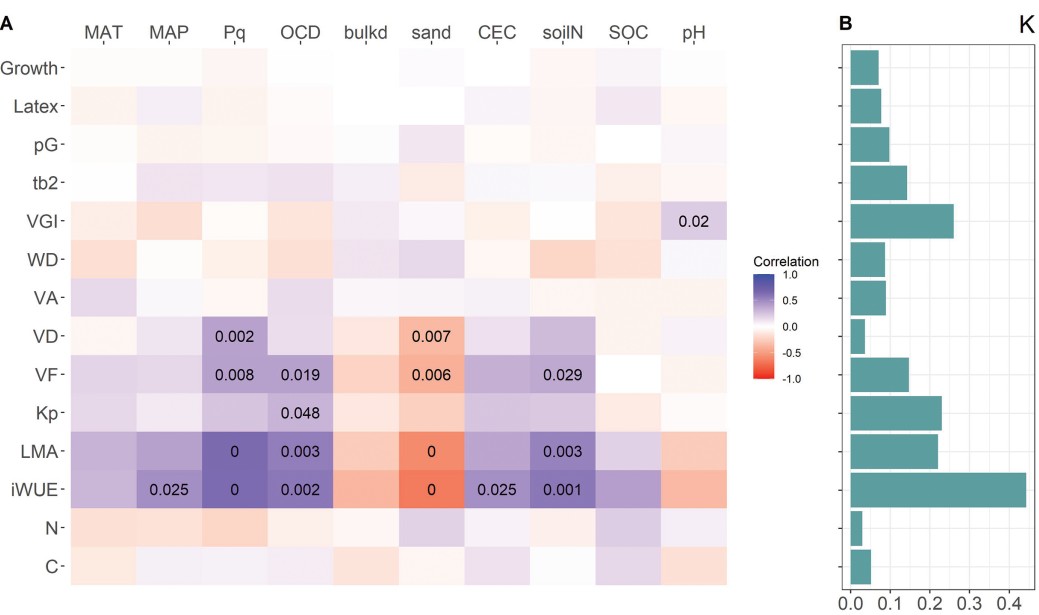

**Figure 3 Heatmap of correlations between resource allocation, trait values, and environmental variables at origin locations (A), and phylogenetic signal (Blomberg's K) of traits (B).** Color code for correlation represents the marginal R of mixed-effect models with location as random variable, numbers are *p*-values < 0.05. For trait abbreviations see Fig. 2, for environmental variables see Table 1.

## Are traits and investment in growth or defense related to genetic adaptations to water or nutrient availability?

In the mixed-effect models with location of origin as random variable, VD, VF, LMA and iWUE were positively related to Pq and negatively to sand content, both affecting water availability, VF, LMA and iWUE also to OCD and soil N (Fig. 3A). Of the wood parameters frequently used as proxies for embolism resistance (VGI and $(t/b)^2$) only VGI was weakly related to soil pH. Also, the first principal component of the standard as well as the phylogenetic PCA was related to Pq, OCD, soil sand and N content (Table 3). Neither investment in growth or latex yield, nor the relative investment in growth (pG) was related to any of the environmental parameters from the locations of origin. With the exception of VGI, traits that showed correlations with specific environmental variables from the location of origin had a high phylogenetic signal (Fig. 3B). Neither growth nor latex yield was related to any environmental variable.

## DISCUSSION

Species differ in their strategies for dealing with enemies and environmental constraints. These strategies can be tuned to and genetically adapt to specific environments, which can be explored by testing if populations from different origins differ in relevant traits when grown under the same environment (*Alberto et al., 2013*). We here used a widespread tropical tree with a well-defined investment in defense to explore the growth-defense trade-off and to test potential adaptations that might affect this trade-off.

**Table 3 Correlations between the first two principal components of the PCA including traits and growth variables and environmental variables from the location of origin.**

| Env.vars | PC1 | | | PC2 | | |
|---|---|---|---|---|---|---|
| | $p$ | $r^2m$ | $r^2c$ | $p$ | $r^2m$ | $r^2c$ |
| **Standard PCA** | | | | | | |
| MAT | 0.285 | 0.057 | 0.654 | 0.095 | 0.051 | 0.109 |
| MAP | 0.065 | 0.152 | 0.645 | 0.154 | 0.034 | 0.104 |
| Pq | **2.10E−05** | 0.467 | 0.548 | 0.768 | 0.001 | 0.112 |
| OCD | **0.011** | 0.236 | 0.613 | **0.041** | 0.067 | 0.111 |
| BulkDens | 0.204 | 0.064 | 0.647 | 0.269 | 0.021 | 0.112 |
| Sand | **1.10E−04** | 0.369 | 0.541 | 0.489 | 0.006 | 0.105 |
| CEC | 0.104 | 0.120 | 0.644 | 0.158 | 0.036 | 0.110 |
| soilN | **0.006** | 0.246 | 0.611 | 0.382 | 0.013 | 0.117 |
| SOC | 0.557 | 0.022 | 0.659 | 0.073 | 0.064 | 0.126 |
| pH | 0.245 | 0.058 | 0.648 | **0.019** | 0.077 | 0.094 |
| **Phylogenetic** | | | | | | |
| MAT | 0.298 | 0.042 | 0.376 | 0.060 | 0.077 | 0.124 |
| MAP | 0.104 | 0.083 | 0.358 | 0.085 | 0.056 | 0.140 |
| Pq | **1.50E−08** | 0.283 | 0.283 | 0.264 | 0.022 | 0.144 |
| OCD | **0.035** | 0.130 | 0.330 | **0.013** | 0.107 | 0.129 |
| BulkDens | 0.250 | 0.042 | 0.370 | 0.569 | 0.007 | 0.149 |
| Sand | **0.001** | 0.203 | 0.256 | 0.148 | 0.031 | 0.123 |
| CEC | 0.185 | 0.063 | 0.355 | 0.240 | 0.031 | 0.143 |
| soilN | **0.023** | 0.134 | 0.330 | 0.218 | 0.029 | 0.148 |
| SOC | 0.623 | 0.012 | 0.382 | 0.301 | 0.030 | 0.166 |
| pH | 0.286 | 0.036 | 0.364 | 0.058 | 0.065 | 0.131 |

Note:
$r^2m$ is the marginal and $r^2c$ the conditional $r^2$ of mixed-effect models with location as random variable. The upper table shows correlations with a standard PCA, the lower table with the phylogenetic PCA. Significant ($p < 0.05$) $p$-values are printed in bold. Environmental variables are MAT, mean annual temperature, MAP, mean annual precipitation, Pq, precipitation in the driest quarter, OCD, soil organic carbon density, BulkDens, soil bulk density, CEC, cation exchange capacity, soilN, soil nitrogen concentration, SOC, soil organic carbon.

## Intraspecific trade-off between growth and defense

Our study found no evidence of a growth-defense trade-off in Brazilian rubber trees originating from climatic and soil gradients. This conclusion contrasts with numerous studies that have found trade-offs between growth and defense traits across species. *Coley (1988)*, *Fine et al. (2006)*, *Lind et al. (2013)*, but studies comparing populations within species are scarce. The amount of phenolic glycosides in leaves of *Populus tremuloides* trades off with the growth rate under high and low competition (*Cope et al., 2021*). However, our findings align with those of *Garcia-Forner et al. (2021)* who found no trade-off between constitutive anatomical resin-based defenses and radial growth in *Pinus pinaster* populations. In nine populations of *Pinus pinaster* planted in a common garden, those from Atlantic climates showed a stronger trade-off between resin duct density and

growth than those from more arid and seasonal climates (*Vázquez-González et al., 2020*). Thus, the relationship between growth and defense traits can vary widely within species (*Hahn & Maron, 2016*), highlighting the importance of considering specific environmental contexts and genetic backgrounds.

## Potential explanations of no trade-off between growth and defense

A growth-defense trade-off may be influenced by a variety of factors such as the specific plant species involved or the type and severity of environmental stressors (*Hahn & Maron, 2016*). Plants may also show variation in growth-defense trade-offs due to variation in nutrient acquisition and allocation (*Van Noordwijk & De Jong, 1986*). Plants allocating more resources to growth can outcompete neighbors for light, nutrients and space, but are more vulnerable to herbivory and other environmental stressors. Conversely, allocating more resources to defense protects a plant from some stresses, but reduces resources available for growth (*Monson et al., 2022*). Thus, the optimal resource allocation strategy may also depend on the plant's environment and life history. The resource availability hypothesis has been invoked to predict intraspecific patterns of plant defense along environmental gradient (*Endara & Coley, 2011*) and proposes that high-resource environments favor a fast-growing plant to allocate resources to growth because tissue damage becomes less costly to fitness under these conditions, whereas under low-resource environments losses to herbivores are more costly to replace, favoring investing in defense (*Coley, Bryant & Chapin, 1985*). Variations in investment should thus depend on resource availability and could be evolutionarily selected and be fixed in a genotype but could also be a phenotypic response to resource availability. For instance, nutrient addition stimulated growth for *Spartina alterniflora* in salt marshes, while decreasing defensive traits, such as fiber and silica content (*Wittyngham, Carey & Johnson, 2023*). *Larrinaga, Sampedro & Zas (2024)* also found that fertilization significantly enhanced resource availability for *Pinus pinaster*, leading to vigorous growth, but reducing resin and phenolic concentrations in the needles. In our study, we tested for genetic adaptations to environments differing in water and N supply, but in trees were grown under uniform and comparatively high-resource conditions (N fertilization and little competition by the spacing between trees) with little apparent herbivore damage. The absence of a growth-defense trade-off under these conditions is similar to a recent study on *Monarda fistulosa* (*Hahn et al., 2021*) and does not support the resource availability hypothesis (*Herms & Mattson, 1992*). However, in relating the trees in a uniform trial environment to the environment of origin of the clones, we only tested for a genetic adjustment to resources and not a possible phenotype-related trade-off. Such a trade-off based on phenotypic adaptations might still be found in low resource environments (*Vázquez-González et al., 2020*), but testing this falls beyond the scope of our current study and the experimental design.

We therefore propose that the absence of a trade-off between growth and defense can be expected for tree populations growing in a resource-rich environment. This may allow to allocate resources to both growth and defense without compromising either (*Van Noordwijk & De Jong, 1986*; *Russo et al., 2022*) as seen in other species such as *Acacia*

(*Ward & Young, 2002*), *Pinus* (*Redmond et al., 2019*), and *Quercus* (*Perkovich & Ward, 2021*). Plants may have evolved mechanisms to efficiently utilize available resources for both growth and defense (*Huot et al., 2014*), which could involve optimizing physiological processes, such as enhanced photosynthetic efficiency, nutrient uptake, and defense responses. Thereby plants can achieve a balanced resource allocation strategy that prioritizes both processes to maximize fitness and overall performance (*Stamp, 2003*; *Huot et al., 2014*; *Monson et al., 2022*). This optimal allocation strategy ensures that plants can grow vigorously while maintaining robust defense mechanisms to withstand stress. This pattern of resource allocation has been explained in the growth-differentiation balance hypothesis (*Herms & Mattson, 1992*; *Stamp, 2003*). Growth generally receives a high allocation priority over defense because growth is a highly resource-demanding process (*Kleczewski, Herms & Bonello, 2010*). When plants are growing in high-resource environments, they can allocate more resources towards growth than is necessary for survival and reproduction. An abundance of carbon resources can result in a surplus that is not being used for growth and is invested in structural or chemical defenses. Such investment in defenses has been described as "resource overflow" of surplus resources not used for growth (*Chapin, Schulze & Mooney, 1990*). These conditions may therefore preclude a trade-off, enabling high growth rates as well as higher latex production in the Brazilian rubber trees grown under high-resource environments at the Thai site. The surplus resources can potentially accumulate reserves that provide safety for responses to future biotic and abiotic stresses (*Babst et al., 2005*; *Monson et al., 2022*). Apart from the resource overflow, a priority to allocation to growth may also better serve for long-term survival. Several studies found that tapped rubber trees accumulate more carbon reserves than untapped trees (*Silpi et al., 2007*; *Chantuma et al., 2009*), so that trees under stress might accumulate reserves, at the expense of growth. This would interfere with the allocation of carbon between growth and defense.

## Associations between functional traits and investment in growth

We had asked whether the allocation of resources towards growth and defense could be linked to other traits associated with stress resistance, even in the absence of a trade-off between growth and defense. The first PCA axis was highly correlated with a suite of leaf functional traits such as LMA, iWUE, and leaf nitrogen content. Growth scaled on the first axis of variation in traits (Fig. 2) and weakly but positively with LMA and iWUE, while it was negatively correlated with leaf nitrogen content. Leaves with higher LMA are generally thicker with more structural compounds (*Wright & Cannon, 2001*). This contrasts with the general association of high LMA and iWUE with a conservative resource use and stress tolerance, rather than high growth rates (*De La Riva et al., 2016*). Water stress-induced increases in LMA are often correlated with increases in iWUE for many growth forms (*Medrano, Flexas & Galmés, 2009*). The relationship between LMA, environment and growth is complex (*Poorter et al., 2009*) and positive (*Wright et al., 2010*) as well as negative (*Poorter et al., 2008*) correlations between LMA and relative growth rates have been reported from tropical trees. In provenance trials of *Populus trichocarpa* no relationship was found between growth and leaf N and weak positive or negative relationships with SLA

(*Mckown et al., 2014*). In the rubber trees we studied, the correlation between growth and LMA and leaf nitrogen content was significant but weak and might be explained by a trade-off in plant resource allocation. Species with high growth rates were found to have a lower proportion of photosynthetic nitrogen in leaves (*Wright & Westoby, 2001*) and several studies found the relationship between mass or area based leaf N, LMA, leaf longevity and growth to be complex and affected by nitrogen or water availability (*Funk, Jones & Lerdau, 2007*). If growth is primarily limited by factors other than nitrogen, high growth rates may result in a dilution of nitrogen content in leaves. In phylogenetic analyses the correlations between traits and investment in growth or latex yield and the overall relationship among traits as seen in the PCA were similar to non-phylogenetic analyses. This means that these correlations are not or not strongly constrained by the genetic relationship among the clones, but not that the differences in traits, growth or latex yield are not genetically controlled and evolving.

## Clinal adaptation to environmental gradients

A previous study of the same trial had reported clinal variation in wood and leaf traits with dry season intensity (*Rungwattana et al., 2018*). We here found mostly the same traits associated also with sand content and soil N concentration and to some extent with OCD. For the present analysis we also included traits that had been found to correlate with embolism resistance (VGI (*Carlquist, 1984*; *Tyree, Davis & Cochard, 1994*) and $(t/b)^2$ (*Hacke et al., 2001*)), but with the exception of a correlation between VGI and soil pH, these did not scale with any variable related to water availability or soils (Fig. 3A). Clines are variations of traits along environmental gradients and when traits measured in a common garden relate to the environment the plants originate from, this points to genetic adaptations to the environment (*O'Neill et al., 2002*; *Woods et al., 2012*; *Pratt & Mooney, 2013*). With the exception of VD, all traits that show clinal variation tend to have strong phylogenetic signals (Fig. 3), and traits with weak phylogenetic signals also showed weak clinal variations, but growth and latex production showed neither. If variation in latex yield is driven by local adaptations, this was not captured by the environmental parameters investigated, but abiotic variables are rather unlikely to be main drivers of defense. Rather variation in latex production can be an adaptation to regional variations in herbivory pressure (*Moreira et al., 2018*; *Morrow, Jaeger & Lindroth, 2022*). Herbivory puts a strong selective pressure on plants, favoring those with effective defense mechanisms (*Boege & Dirzo, 2004*; *Agrawal & Fishbein, 2006*; *Züst et al., 2012*; *Poelman & Kessler, 2016*), the importance of which can also interact with variations in resource availability. Latex production is known to be phenotypically adjusted as a response to herbivory in *Asclepias* and *Hevea* (*Agrawal & Konno, 2009*), and knowledge of this is used to increase latex production by applying jasmonic acid, a phytohormone involved in defense (*Hao & Wu, 2000*). While the phylogenetic signal was low (Fig. 3B), latex yield is certainly also genetically controlled and rubber trees can be selectively bred for high yield (*Cheng et al., 2023*). Given that our hypotheses on factors driving latex production and a potential growth-defense trade-off were not confirmed, the question of the evolutionary drivers remains unanswered.

## CONCLUSIONS

We looked for a growth-defense trade-off and tested if the resource availability hypothesis applies at the intra-specific level in a provenance trial that tested for genetic rather than phenotypic variation. Trees growing in a high-resource environment at the Thai site showed no trade-off between growth and defense. While wood and leaf traits showed clinal adaptations to the Brazilian environment the clones originate from, investment in defense, although very variable, remained unexplained by any of the factors explored. If these variations reflect adaptations to variable herbivory pressure or indirect effects remains unknown. *Hevea brasiliensis* is an interesting model species to explore underlying mechanisms of defense evolution, which is also relevant for breeding high yield rubber trees. The underlying causes for the high variation in latex production in one of the commercially most important tree species remains unexplained.

## ACKNOWLEDGEMENTS

For wood sample collection, we thank the Rubber Research Institute of Thailand. We thank Susanne Scheffknecht for helping VGI calculation and Klara Voggeneder for soil variables image analysis. We thank Phillippe Thaler and Daniel Epron for their contribution of leaf traits and insightful comments.

### Funding

This work was financially supported by Office of the Permanent Secretary, Ministry of Higher Education, Science, Research and Innovation (RGNS 64-025). Fieldwork of Peter Hietz and wood anatomical analyses by Kanin Rungwattana were supported by the Austrian Federal Ministry of Science, Research, and Economy within the framework of the ASEA UNINET. Kanin Rungwattana was supported a grant from the ASEA UNINET (ASEA 2021-2022/BOKU/5) and the Faculty of Science, Kasetsart University (PRF-P 3/2563, PRF 3II/2564, and International SciKU Branding). There was no additional external funding received for this study. The funders had no role in study design, data collection and analysis, decision to publish, or preparation of the manuscript.

### Grant Disclosures

The following grant information was disclosed by the authors:
Permanent Secretary, Ministry of Higher Education, Science, Research and Innovation: RGNS 64-025.
ASEA UNINET: ASEA 2021-2022/BOKU/5.
Faculty of Science, Kasetsart University: PRF-P 3/2563, PRF 3II/2564, and International SciKU Branding.

### Competing Interests

The authors declare that they have no competing interests.

## Author Contributions

- Kanin Rungwattana conceived and designed the experiments, performed the experiments, analyzed the data, prepared figures and/or tables, authored or reviewed drafts of the article, and approved the final draft.
- Poonpipope Kasemsap analyzed the data, authored or reviewed drafts of the article, and approved the final draft.
- Thitaporn Phumichai analyzed the data, authored or reviewed drafts of the article, and approved the final draft.
- Ratchanee Rattanawong analyzed the data, authored or reviewed drafts of the article, and approved the final draft.
- Peter Hietz conceived and designed the experiments, performed the experiments, analyzed the data, prepared figures and/or tables, authored or reviewed drafts of the article, and approved the final draft.

## Data Availability

The data that support the findings of this study are available in the Dryad Digital Repository: Rungwattana, Kanin et al. (2024). Allocation to growth and defense in Hevea [Dataset]. Dryad. https://doi.org/10.5061/dryad.6q573n65b.

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
