# Peer review of "Testing intra-species variation in allocation to growth and defense in rubber tree (Hevea brasiliensis)"

_PeerJ, doi:10.7717/peerj.17877_

## Round 0.1 · original submission · Major Revisions

Dear authors,

Your manuscript has been reviewed by two experts. Both found value in your research and are supportive of your work. However, both reviewers identified issues that need to be addressed before the manuscript can be reassessed. In particular, both found that the analyses need improvements, in terms of methods used as well as in terms of matching the experimental design and conclusion. I agree with their assessment.

Therefore, my decision is Major revision.

If you decide to submit a revision, please provide a point-by-point response to the reviews. Please note that the revised manuscript will undergo review and acceptance cannot be guaranteed at this stage.

I am looking forward to reading the revised manuscript.

Best regards

Yann Salmon

Reviewer 1 ·

Basic reporting

The study aims to test if a potential trade-off between growth and defence exists in different clone of rubber tree and what traits were associated to these parameters. The authors reported that no trade-off was found; growth was weakly correlated to the plant traits and defence was not correlated to any traits or climate factor at all. The authors also proposed potential reason. The paper is properly structured and all. Raw data has been supplied. The authors provide sufficient background of the study and listed several hypotheses and explained well. But the introduction could be still improved. It would be nice if the author points out clearly which hypothesis they are aiming to test and how to link it with a testable question.

Experimental design

The study is original. Though a part of data has already been published in other two paper, the author compiled old data with newly measured parameters and conducted analysis from another point of view. The paper fills the gap on growth-defence trade-off at intra-specific level. The experiment is well-designed with enough replicate to answer the question.
I have some comments on the details of question and provided in the method and material:
The questions are generally will formed, but question 4 (line 140) is not straightforward, please reframe it. “Sink limitation” appears only twice in the paper. Does “Sink limitation hypothesis” here refers to line 86 “Consequently, plants facing sink limitation due to stress tend to display higher resistance to external constraints compared to plants that encounter no environmental limitations in their growth (Herms & Mattson, 1992)” ? If that is the case, it would be nice to explain and give the name in the previous paragraph. Additionally, if this is one of the questions you want to address in this study, it would be nice to discuss it properly and make a conclusion after.

The first one is the site and sampling description relied too much on other published paper. I understand that the descriptions (table and the map) of parent material original locations have been published and the authors try to avoid repeat the same things. However, it is inconvenient for readers to search for 2 more papers to understand the basic information, e.g. the number of the locations. Additionally, in the discussion the author mentioned several times that it may because clones were all planted without stress, a trade-off of growth and defence cannot be found. It would be clearer to at least have a table or graph to show the climate of the common garden with the compares of the original location.
The second one is on data analysis. The author paid attention on choosing models considering the data is not dependent. But models in method and corresponding results are somehow confusing.
The author used mixed effects model on growth or latex yields production response to climate or soil environment factor (line 235), and this is the only time author mentioned mixed effect model in the method. The result of this model is not mentioned in the result section, instead the author mentioned the result of latex response to tree size and tree growth (line 255), which is not mentioned in the method. The authors mentioned regression between PC1/PC2 to bioclimate factor, without mentioning random factors. But in Table 1, the authors said it has been down by mixed effect model. Figure 4 showed a mixed effect model result with Growth, or latex or functional traits as response variable. But in the method, the models are not mentioned.
The author used phylogenetic generalized linear models to test the relationship between growth/latex and traits, but the result is never mentioned.
There are also some other minor comments on the details of method:
Line 159: I suppose it is 500 g N:P:K (30:5:18) fertilizer per tree. Please specify it.
Line 162: It would be nice to describe the range of “precipitation of driest quarter” for original location as well.
Line 186: Please write the year when wood samples were collected. It seems that they are collected in a different year from leaf samples according to Rungwattana et al. 2018
Line 214: The word “Biomass Growth” and “Growth” (biomass Growth/D), “Latex yield” and “Latex” (latex yield per tapping/D) are confusing. The author defined “Growth” and “Latex” at line 214. However later in the paper, the author used “growth and latex”, sometimes use growth and latex yield (e.g. line 235)
Line 235 and line 243, please cite the package mixed effect model and pagel’s lambda. Perhaps it is better to call it “mixed-effects model”.
Line 237 Please reframe the sentence. It is difficult to follow.
Figure 3, the label of VGI overlap y-axis.

Validity of the findings

The study was based on 150 clones with replicates, The conclusions are well stated and well supported by the result. All the figures provide clear information to support the conclusions and show that the results appear statistically correct. The authors did not over interpolate result. There are some mismatches between the result provided and the data analysis method, please see experiment design.
In general, I found the discussion could be still improved.
Line 289 It seems that this is one of the main conclusions of the study. The author did not find any trad-off, it may be better to start with conclusion (not trade-off).
Line 343, the subtitle of this section is “Associations between the investment in growth, defense, and stress resistance”. The authors discussed the relation between growth and some functional traits, but did not discuss defense clearly. I recommend to either change the title or add more efforts to discuss defense or stress.

Additional comments

The paper is well presented. I am very supportive of the study which claims non-significant result. there are still some minor comments on some ambiguity of language and typos:
Line 28: Please specify which hypothesis. In the introduction, the author mentioned several.
Line 48 …, and water are often limiting in natural habitats… The word ‘limiting’ should be limited.
Line 52: I understand that it is common-sense that a trade-off exists either because of carbon or nutrient. However, the logic link between this sentence and the previous one is not clear.
Line 253, in the method part (line 214) the author mentioned that growth and latex were corrected by diameter, but here the author mentioned result with two parameters corrected by circumstance. It might give the similar conclusion but it is better to be consistent.
Line 277 it would be nice to have a separate paragraph for the result of phylogenetic related analysis. Additionally, where is the result of mixed effects model?

·

Basic reporting

There are minor revisions to references. The addition of newer references would enhance the validity of the paper.

Experimental design

The design was well-thought out and implemented. However, the statistical analysis is done poorly and should be reassessed.

Validity of the findings

No comment.

Additional comments

In this manuscript, the authors sought to understand the relationship between nutrient availability and investment in the growth-defense trade-off. The authors used clones of rubber trees to understand the within species variation of patterns in growth and defense trade-offs. The authors found limited support for any patterns of growth vs defense investment correlating with environmental factors. This manuscript is of great value to the field and would be of interest to many readers of Peerj. I enjoyed reading the introduction and felt the article was promising. However, as I continued, there are several issues in the methods and the statistical interpretations that make it hard for me to evaluate the discussion portion of this paper. The issue is major, and should be addressed before publication. However, it is a matter of redoing the statistical analysis and re-evaluating their conclusions. This could be accomplished by the authors and resubmitted.

---

## Round 0.2 · Minor Revisions

Dear authors.

Both reviewers are pleased with the improvement made to the manuscript and I agree that only very minor improvements are needed now. Please, address the minor issues mentioned by reviewer 1. I also encourage you to consider the sentence by reviewer 2 regarding the correlation coefficient.

Additionally:

1) it would be useful to add letters to the different panels (e.g. Fig. 1).

2) the regression is Fig.1 Biomass growth is not a linear one. Look at the residual: the model underestimate values at low and high diameter and overestimate at medium diameter.
This will not change the interpretation of the result, but it should be addressed nonetheless. Please also check if the linear model works for latex yield etc.

3) PeerJ does not provide proof-reading, thus you need to correct any missing punctuation marks or typos.
Thank you

Reviewer 1 ·

Basic reporting

This perspective manuscript is the revised version of a manuscript which I also reviewed. The authors investigated if the defence of rubber tree (latex yield) shows a trade off with tree growth, however the author did not find any trade-off between defend and growth.
The manuscript has been significantly revised, and I admire the new version with more important details. I agree with the points made in rebuttal letter as well.

Experimental design

The experiment was well designed. I appreciated the new version of the data analysis section, which gives a much clearer indication on why and how did the authors perform these tests.

Validity of the findings

No comment.

Additional comments

There might be a few missing punctuation marks or a typos in some places that I am sure can be dealt with during the proof-reading stage.
Line 55: Such a trade - off will result in trees that grow quickly tend to have weak defenses, while those with strong defenses tend to grow slowly (Fichtner et al., 2017; Z¸st & Agrawal, 2017; Zhu et al., 2018).
…tend.. to tending
Line 67: remove “those of”
Line 96 “source – sink” to source-sink
Line 264 ” Bloomberg's” to Blomberg’s
Line 298 “was marginally related to”
Figure 3: What is the maxim value of Blomberg's K
Table 1, Line 4: “N: number of clones analysed”. The number of the clones is not in the table. I suppose “N” in the table refers to soilN in the footnote (Line 6). Please verify it.

·

Basic reporting

The manuscript is much improved. The authors did a great job addressing and making appropriate changes. They have included newer citations along with primary works that help support their interpretations of the data.

Experimental design

The authors have improved their explanation of the experimental design.

Validity of the findings

While I do not agree with their interpretations of the correlation coefficients, statistical values are provided and readers can interpret them as they please.

Additional comments

I am happy with the updates and changes to the paper. I think it is a fascinating study and is well written at this point.

---

## Round 0.3 · accepted · Accept

Dear authors,

Thank you for addressing the reviewer’s comments. Congratulations, your manuscript is now suitable for publication.